# Psychosocial Risks among Quebec Healthcare Workers during the COVID-19 Pandemic: A Social Media Analysis

**DOI:** 10.3390/ijerph20126116

**Published:** 2023-06-13

**Authors:** Maryline Vivion, Nathalie Jauvin, Nektaria Nicolakakis, Mariève Pelletier, Marie-Claude Letellier, Caroline Biron

**Affiliations:** 1Department of Scientific Valorization and Quality, Institut National de Santé Publique du Québec (INSPQ), Quebec, QC G1V 5B3, Canada; 2CHU de Québec-Université Laval Research Center, Quebec, QC G1V 4G2, Canada; 3Department of Environmental and Occupational Health and Toxicology, Institut National de Santé Publique du Québec (INSPQ), Quebec, QC G1V 5B3, Canada; nathalie.jauvin@inspq.qc.ca; 4Department of Environmental and Occupational Health and Toxicology, Institut National de Santé Publique du Québec (INSPQ), Montreal, QC H2P 1E2, Canada; nektaria.nicolakakis@inspq.qc.ca; 5Guidance and Counseling School, Université Laval, Quebec, QC G1V 0A6, Canada; marieve.pelletier@fse.ulaval.ca; 6Department of Public Health of Gaspésie-Îles-de-la-Madeleine, Gaspe, QC G4X 1A9, Canada; marie-claude.letellier.med@ssss.gouv.qc.ca; 7Department of Management, Faculty of Business & Administration, VITAM—Research Center for Sustainable Health, Université Laval, Quebec, QC G1V 0A6, Canada; caroline.biron@fsa.ulaval.ca

**Keywords:** psychosocial work environment, work-related psychosocial risk, healthcare worker, social media, COVID-19, qualitative research, Quebec

## Abstract

During the COVID-19 pandemic, healthcare workers (HCWs) were at high risk of exposure to the SARS-CoV-2 virus and to work-related psychosocial risks, such as high psychological demands, low social support at work and low recognition. Because these factors are known to be detrimental to health, their detection and mitigation was essential to protect the healthcare workforce during the pandemic, when this study was initiated. Therefore, using Facebook monitoring, this study aims to identify the psychosocial risk factors to which HCWs in Quebec, Canada reported being exposed at work during the first and second pandemic waves. In this study, HCWs mainly refer to nurses, respiratory therapists, beneficiary attendants and technicians (doctors, managers and heads of healthcare establishments were deemed to be less likely to have expressed work-related concerns on the social media platforms explored). A qualitative exploratory research based on passive analysis of Facebook pages from three different unions was conducted. For each Facebook page, automatic data extraction was followed by and completed through manual extraction. Posts and comments were submitted to undergo thematic content analysis allowing main coded themes to emerge based on known theoretical frameworks of the psychosocial work environment. In total, 3796 Facebook posts and comments were analyzed. HCWs reported a variety of psychosocial work exposures, the most recurrent of which were high workload (including high emotional demands), lack of recognition and perceived injustice, followed by low workplace social support and work–life conflicts. Social media monitoring was a useful approach for documenting the psychosocial work environment during the COVID-19 crisis and could be a useful means of identifying potential targets for preventive interventions in future sanitary crises or in a context of major reforms or restructuring.

## 1. Introduction

### 1.1. Work-Related Psychosocial Risks during the Pandemic

Healthcare workers (HCWs), such as physicians, nurses, nursing assistants, patient healthcare support personnel and beneficiary attendants have been at high risk of exposure to SARS-CoV-2 throughout the pandemic. In the province of Quebec, Canada, their risk of infection was 10 times higher than that of the general working-age population during the first pandemic wave and four times higher during the second wave [1,2]. This significant exposure to the virus, combined with challenging working conditions, posed a threat to HCWs’ physical and psychological health. The scientific literature suggests that HCWs are, from the outset, a population of workers particularly at risk of developing mental health issues [3,4,5], and that this risk tends to be accentuated during epidemics and pandemics [6,7,8,9,10]. Indeed, several studies have shown, on a global scale, that the COVID-19 pandemic had a negative impact on HCWs’ mental health [1,2,11,12,13,14,15,16,17]. Factors associated with increased mental health problems during the pandemic included work overload, lack of resources and equipment, lack of organizational support, lack of training and preparation, fear of being contaminated or of contaminating users, loss of confidence in government authority when its discourse did not correspond to the realities on the ground, and value conflicts or moral dilemmas triggered when faced with certain choices (e.g., caring for patients while risking one’s own health and that of one’s family, having to choose who receives care when resources are limited, etc.) [7,18,19]. 

Some of these risks faced by HCWs during COVID-19 were more specifically related to the pandemic context (e.g., fear of being infected or of spreading the disease), but most are recognized work-related psychosocial risks whose effects had been previously widely documented in workers [1,2,20,21,22,23,24,25,26]. According to the Quebec Public Health Institute (INSPQ), psychosocial risks in the workplace are defined as factors related to work organization, management practices, employment conditions and social relations at work, all of which increase the likelihood of adverse effects on the physical and psychological health of workers exposed [27]. Workplace psychosocial risk factors such as high psychological demands, low support at work and low recognition are significant determinants of psychological distress, depression, cardiovascular and musculoskeletal outcomes [20,22,24,25,28]. Value conflicts and difficulties experienced while having to balance work and personal life are also increasingly recognized components of the psychosocial work environment associated with psychological distress [29,30]. During the COVID-19 pandemic, care providers faced multiple moral conflicts in a context of stretched healthcare systems and strict infection and prevention control rules [31]. Balancing work and personal life such as home schooling was conceivably more difficult with the additional workload caused by the pandemic. 

The identification of workplace psychosocial risks is a prerequisite for the design and implementation of preventive interventions to protect HCWs’ mental health [32,33]. Therefore, the objective of this study was to identify workplace psychosocial risk factors experienced by HCWs during the COVID-19 pandemic by the use of social media. This innovative approach offers us the possibility to study psychosocial work exposure without directly soliciting HCWs. Indeed, risk assessment by way of a questionnaire or through focus groups can be burdensome, if not impossible, during a health emergency, especially for HCWs working at the heart of the crisis, prompting us to consider this alternative method. Although applied in a pandemic context, this method can also be used in the context of other crises or major changes that impact the healthcare system (reforms, restructuring), allowing for the study of work-related psychosocial risks when other means of risk assessment are impractical or unfeasible. 

### 1.2. A Health System Already Vulnerable

In order to better understand the prevailing context in the Quebec healthcare system at the time of the study, we have provided below a brief historical account of this system’s most recent major reform, as well as key pandemic events. These are also summarized in Figure 1.

In 2015, Bill 10, which aimed to fundamentally reform the Quebec healthcare system, was passed. Commonly referred to as “Barrette’s reform” (named after the Health Minister who implemented it), it led to the abolition of regional health and social services agencies and the creation of large centralized Integrated Health and Social Services Centres (CISSS) and Integrated University Health and Social Services Centres (CIUSSS) [35] in each of Quebec’s 18 administrative regions. Each centre managed all of the public hospital centres, residential and long-term care centres, rehabilitation centres, child and youth protection centres and local community service centres in their region. Although the aim was to improve governance efficiency, evaluations of this reform indicated that it resulted in weakening several services [36], and often eroded employees’ sense of collective belonging to their workplace. This was in part due to the reduction of the number of middle managers and the ensuing loss of proximity of upper management to local needs and concerns [37].

With the arrival of the pandemic, some of the issues generated with the reform and the existing HCWs shortage were amplified. The Quebec Minister responsible for Seniors and Caregivers had indicated in April 2019 that there was a shortage of 33,036 beneficiary attendants, 23,963 nurses, 4068 health and social services assistants, 656 institutional pharmacists and 895 psychologists [38]. Thus, when the health emergency was declared in Quebec on 13 March 2020, the healthcare network was already significantly understaffed and operating under challenging working conditions [39,40,41]. At the beginning of April 2020, the lack of personal protective equipment (PPE) was reported [42] as COVID-19 outbreaks struck long-term care facilities, leading to a substantial number of patient deaths. To counter this dramatic situation, medical specialists and the Red Cross, as well as the Canadian army, were called into the province as reinforcements. 

After the pandemic was declared as an international emergency on 21 March 2020, a ministerial order was issued to “allow employers to have the necessary human resources” in the healthcare network, allowing them to move employees between establishments. The ministerial order also suspended HCWs’ vacation time and authorized employers to modify their working time arrangements (e.g., the four-day work week, vacations), in addition to resorting to the use of the existing compulsory overtime (CO) measure to compensate for the lack of resources [43]. In May 2020, a new directive allowed for mandatory quarantine requirements for HCWs to be lifted to avoid service interruption.

In May of 2020, to address the staffing shortage, the Quebec Premier announced the creation of a training program to recruit 10,000 beneficiary attendants. This training was paid at $21 per hour upon successful completion, with an increase to an hourly rate of $26 “via bonuses” or the “collective agreement”. Negotiations were also initiated with HCW unions regarding COVID-19 bonuses, staff recognition, and reductions in patient-to-staff ratios, culminating in the granting by the government of a 4% bonus to HCWs that same month [37].

The second pandemic wave began at the end of August 2020. At that time, the staffing shortage had reached acute levels, with resignations and HCWs work absences related to COVID-19 or mental health issues up by 43% among nurses since the beginning of the pandemic [44], and PPE was also still lacking in some healthcare centres [42]. 

This context suggested that HCWs encountered work-related psychosocial risks that needed to be addressed. Therefore, our team initiated a participatory research project aimed at gathering knowledge on organizational strategies to protect HCWs’ mental health during the COVID-19 pandemic, with the ultimate goal of developing a tool to support healthcare establishments in their mental health prevention efforts. This project integrated three complementary approaches to gathering knowledge. It included a systematic literature review evaluating the effectiveness of interventions targeting work organization or the psychosocial work environment to protect HCWs’ mental health during epidemics [45,46]. It also included an online survey about organizational measures implemented in healthcare establishments to mitigate psychosocial work exposure during the pandemic, as reported by human resources staff. This paper reports on the third approach used in the larger research project, that is the identification of workplace psychosocial risks being experienced by HCWs during the COVID-19 pandemic, allowing us to align development of the tool with workers’ concerns as expressed on social media. 

## 2. Materials and Methods

### 2.1. Design 

Qualitative exploratory research based on passive analysis of Facebook content was conducted [47]. Three Facebook pages from different healthcare unions were selected as they did not require a subscription and allowed us to reach a variety of health professionals, including nurses, respiratory therapists, beneficiary attendants and technicians. Altogether, the approximate number of subscribers for these three groups was 153,000. All health professionals who were members of the group could post, comment or react to posts and comments. 

### 2.2. Data Extraction

For each Facebook page, data were extracted by two methods. First, data extraction using NCapture, a software from NVivo, was performed [48], allowing Facebook page content to be automatically extracted. However, as the content that is captured tends to be what is most recently posted, we followed up with a manual extraction approach, using a grid to counter this software limitation and ensure that the entire study period was covered. All posts and comments referring to psychosocial risk factors were included in a data extraction file. The extraction file contained the following data: date, post or comments, and work-related psychosocial risk (high psychological demands, low recognition and perceived injustice, low workplace social support from supervisors and from colleagues, work-life conflict and lack of decisional autonomy). Posts and comments referring to union agendas and logistics were excluded, such as calls for a strike or the negotiation of a new agreement. The extraction periods were 1 March to 1 July 2020 to cover the first pandemic wave and 1 September to 17 December 2020 to cover a large part of the second wave. In total, 3796 posts and comments were analyzed, including 1994 posts for the first wave and 1802 posts for the second wave.

### 2.3. Data Analysis 

A thematic content analysis was conducted on posts and comments. This method refers to the transposition of a given corpus into a number of themes representative of the content being analyzed, in relation to the research focus [49]. Data were organized into main coded themes, according to the most influential theorical frameworks of the psychosocial work environment and its dimensions [21,23,50]. The main coded themes consisted of high workload, low recognition and perceived injustice, low social support from colleagues or supervisors, work-life conflicts and lack of decisional autonomy. The coding of the first posts and comments was presented to the research team for validation prior to subsequent coding. It is important to note that although the coding was guided by a set theoretical framework, new themes could be added over the course of the coding, as was the case with the “unsafe working conditions” theme (see results). Posts and comments could also be associated with more than one theme. Definitions and themes associated with the main psychosocial risk factors are in Table 1.

Due to the large amount of data, extraction and coding were carried out by members of the research team (M.V and N.J) and medical students as part of their training. All students were supervised by the research team and received training on qualitative data analysis. Thus, a total of nine people participated in the extraction and coding of the content. To ensure consistent coding, each data set was coded by two different individuals. The data were first pre-coded in an Excel file then imported in Nvivo to be coded by a different person, and agreement was assessed by the students. Finally, the principal investigator (M.V) conducted a series of regular spot checks to validate the coding done by the students. Ambiguous posts and comments were discussed between the first and second author to reach consensus on their meaning and coding. Quotes were translated into English by the research team and revised by a professional translator.

## 3. Results

The analysis allowed us to identify the main aforementioned workplace psychosocial risk factors, as well as manifestations of emerging risks, illustrated below by selected excerpts. The most recurrent psychosocial risks identified were high workload including high emotional demands (*n* = 1902), lack of recognition and perceived injustice (*n* = 1401), followed by low workplace social support (*n* = 978), work–family conflict (*n* = 201) and lack of decisional autonomy (*n* = 84). 

### 3.1. High Workload 

High workload emerged as one of the main work-related psychosocial risks. It was expressed through comments related to highly restrictive work hours and schedules, involving doubts and uncertainties about scheduling, mandatory overtime or the cancellation of vacation time, as well as the impossibility of taking a break: 


*“Over 8 days, 5 days of mandatory overtime!!! It makes absolutely no sense, going to work without knowing if you will be able to leave at the end of the shift.”*

*(Second wave)*


High workload was also linked to the lack of staff, insufficient staff-to-patient ratios, lack of relief, welcoming new staff, staff reassignment or staff coming from external placement agencies: 


*“We have extra work but not the extra staff. We’ll end up just putting out fires.”*

*(First wave)*


The unpredictability of the work and the burden of numerous and complex tasks were also components of high workload: 


*“The workload is excessive! Patients are coming in sicker and sicker with complex care complications! We need to hold a heavy load nurse patient ratio! I am discouraged and I only just started my shift!”*

*(Second wave)*


### 3.2. High Emotional Demand 

In addition to a high workload, emotional demands also emerged from the analyses. Value conflicts associated with a sense of not being able to get the work done or endangering patients contributed to this emotional demand:


*“I have 47 patients and it’s impossible to provide safe and humane care with two teams. Now the number has gone up to 72 patients. So where is the staff to help us?”*

*(First wave)*


The fear of becoming a vector for the virus and infecting patients or loved ones also gave rise to conflicts in terms of values and contributed to high emotional demands: 


*“Being a nurse and having the obligation to go to work to provide care … our children do not have to suffer contamination because of our professional situation … in any case, I will protect my children from this proximity which will make them suffer from this virus and its really unpleasant symptoms”*

*(first wave)*


In the comments analyzed, the fear of becoming a vector was mostly tied to unsafe working conditions, such as the real state of disrepair of the premises, the lack of air-conditioning, the size and layout of the premises, which made compliance with health guidelines extremely challenging, the lack of PPE and inadequate management of uniforms. This fear was sometimes compounded by the lack of instructions or training in hygiene measures, the difficulty and ultimate failure to implement hygiene measures, trouble accessing SARS-CoV-2 testing, the fact that some employees who tested positive for SARS-CoV-2 still had to go to work or return to work before being fully recovered, as well as staff traveling between facilities and between “hot” and “cold” zones (zones with confirmed COVID-19 patients and zones with uninfected patients, respectively), which was reported: 


*[…] “Staff moves easily from red zones to green zones! Agencies from Montreal are brought in (in the first wave, an agency from Montreal had contaminated the seniors’ residence next door)” […]*

*(second wave)*


It was also very clear that high emotional demands were placed on staff through the compassion they expressed towards patients who were suffering or dying.


*“I stroked the hair and held the hand of a patient while her husband and daughter looked at her from the other side of the window that separated her from them … I told her again and again that she was not alone, that she could go gently, that her loved ones loved her, that they were close by … but too far away to hug her one last time … it is a sadness beyond words. I saw parents looking at their children behind the window, without being able to say goodbye. Spouses behind the same window silently crying for their love, without being able to kiss them one last time, to say I love you before they pass away.”*

*(First wave)*


Lastly, emotional demands frequently materialized as secondary trauma exposure: COVID-19-related deaths of healthcare professionals led to many Facebook messages of sympathy for families and loved ones while generating concern, exasperation and even anger:


*“The death of a colleague from COVID-19. Another human being with a big heart dead in combat. I can’t take news like this anymore!!! I am a PAB (beneficiary attendant) too, about the same age (49) as him; I also work in a hospital. Where is the protection, are the equipment and procedures sufficient to protect healthy employees … Another falls in combat, one too many RIP to this angel of health. My sincere condolences to the family. Wishing that all are not forgotten…”*

*(first wave)*


### 3.3. Lack of Recognition and Perceived Injustice

Another significant aspect of the psychosocial work environment that emerged as a risk factor in the Facebook content analysis was the lack of workplace recognition and a related theme, which is that of perceived injustice. Recognition issues included feelings of unfairness and inequitable treatment, as it was reported that HCWs’ recognition varied by profession. For example, nurses and beneficiary attendants got more media coverage and bonuses, while many other professions such as radiology technologists, respiratory therapists, home care workers, specialized educators and clerical staff did not receive such recognition. Most of the comments were aimed at claiming recognition for all of the professionals involved, described by some as “shadow workers.”


*“Why aren’t respiratory therapists ever named anywhere! With a respiratory virus, all the more reason to put a spotlight on our profession! As respiratory therapists, we exist and we are on the front lines of danger in terms of being vulnerable to infection! Intubation goes straight to the heart of the matter!”*

*(First wave)*


Lack of recognition also came across as a lack of respect, a feeling that was mentioned in several comments with expressions like “being sacrificed”, “cannon fodder”, “slaves”, “being laughed at in the face”, “being treated like sh*t”, “being taken for an idiot”. Some also questioned the “guardian angel” reference used during the pandemic. 


*“If instead of calling us guardian angels, the government called us EXPENDABLE 
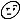
 at least then, it would be more honest of them seeing as how important they consider us to be.”*

*(First wave)*


The fear of speaking out regarding inadequate working conditions was also reported. Some testified to a certain omerta or a “Code of Silence” in their institution. It should be noted that the government had set up an e-mail address to give workers the opportunity to express themselves. However, this initiative was criticized.

Issues with workplace recognition in the form of salary were also expressed, namely as they pertained to the awarding of bonuses. Since bonuses were only intended for full-time staff, this created a sense of injustice for employees not working full-time. This feeling of injustice was also highlighted among beneficiary attendants: the fact that a bonus was included in the salaries of newly hired attendants had not been made clear, which led to a feeling of injustice and non-recognition of existing staff members’ undeniable experience. Thus, governmental interventions intended to fill positions and create attractive working conditions simultaneously created feelings of injustice and a lack of recognition. It should be noted that this was only mentioned on the Facebook page dedicated to the beneficiary attendants’ union:


*“Can someone explain to me how the government is promising full-time jobs and $49,000 salaries to new PABs (beneficiary attendants), even though the [long-term care home] need them, while those already in place do not have these conditions? Is there something I am missing, including respect for the PABs who are already working?”*

*(First wave)*


The discussion surrounding bonuses also generated debates on the establishment of fair wages based on the usefulness of work, the essential nature of the job or acquired diploma. Financial issues were also expressed regarding compensation in the case of illness or inability to return to work or for families in the case of death:


*“I don’t know what the government is waiting for to take action and enhance the value of jobs in the health care sector. The task at hand is huge and we deserve safe working conditions, at the very least.”*

*(Second wave)*


### 3.4. Little Support from Supervisors and Colleagues

Little social support or lack thereof was also a significant psychosocial risk that emerged from the Facebook content analysis. The theme of social support was related to inadequate support from supervisors and from higher authorities, as well as from unions, colleagues or the public.

Lack of support from supervisors was most frequently mentioned and expressed in several ways, including the denouncing of management practices such as lack of listening or the use of threats:


*“It’s unfortunate that we have to fight to get equipment, for our pregnant workers to be able to work in a safe environment, to procure uniforms, but this is the reality we are faced with.”*

*(First wave)*


Lack of emotional support from supervisors was also expressed:


*“WoW I am not at all surprised!!! With them you have to hide everything to look good (…) for too long our complaints are not being heard! Arriving at work, running into your supervisor who doesn’t even say hello!”*

*(First wave)*


Posts and comments also highlighted inadequate informational support from supervisors, expressed as lack of information, contradictory information, or transparency issues: 


*“I deplore our supervisors’ lack of transparency regarding COVID. We are not advised that certain colleagues have COVID and several nurses have it […]. We as the agents are practically not protected we ask for the necessary cleaning gel, virox wipes, etc. We are told that it is for the home care staff to clean their equipment, etc. We are in direct contact with all home care staff. I believe that we have the right to know who is infected in order to protect ourselves. It’s top secret.”*

*(First wave)*


Lack of support from the highest authorities was linked to discrepancies between government declarations and the reality on the ground:


*Legault (the Premier) lied to nurses and the public. He went from reassuring discourse, in which he said he had enough equipment to get through the crisis, to “I will tell you the truth … We have some equipment to last for 3 to 7 days … Nurses need to disinfect the N95 masks.”*

*(First wave)*


Low support from colleagues was evident through comments denouncing the absence of mutual aid, or discomfort with actions taken by the union: 


*“At the hospital we are all a team and we all have our role to play! The important thing is to support each other. We know our worth 
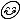
 because I’ve noticed there are some whose nerves are frayed and who are starting to lose patience with each other. We must not destroy each other but rather support each other despite our fears and fatigue (…) otherwise we will all be exhausted and our health will take a hit!”*

*(First wave)*


### 3.5. Work–Family Conflict

Many HCWs expressed much difficulty balancing work and personal responsibilities, such as childcare, home schooling and education, particularly those with young children:


*“Being a single mom, it won’t be easy during this period when daycares are closed. Although we are passionate about our profession, this time is causing us a real headache because the planning issue is a logistical puzzle.”*


### 3.6. Lack of Decisional Autonomy

A less-mentioned aspect of the psychosocial work environment was that of decisional autonomy. When raised, it was mainly in terms of the lack of consultation with employees and the insufficient leeway:


*“I also think that the real problem, apart from the Barrette reform, is the bad management at the CISSS, which is taking employees for fools since long ago. They don’t give any importance or credibility to employees’ suggestions.”*

*(Second wave)*


## 4. Discussion

### 4.1. Main Findings

The results of this qualitative study highlighted several workplace-related psychosocial risk factors reported by HCWs on social media during the first and second COVID-19 pandemic waves in Quebec, Canada. These risks were shown to be detrimental to HCWs mental health during the pandemic in several studies [1,2,53,54,55,56]. This study also highlighted the usefulness of social media analysis as an alternative method of gaining an understanding into workers’ realities when other means of data collection were not feasible, or very low response rates would be expected given the excessive overload faced by the HCWs. Facebook content analysis is an innovative approach that allows us to identify different forms of workplace psychosocial exposure without directly soliciting the healthcare workforce. The free expression of HCWs on social media can provide valuable insight into the complexities of their working conditions and bring attention to the issues that they deem important, which may otherwise be incompletely documented—for example, if reference is made to survey data alone. 

High workload was among the main psychosocial risks expressed by HCWs. It refers to high physical and psychological demands at work, time pressure, frequent interruptions and contradictory demands [21]. Previous studies have shown that these stressors could lead to adverse mental, cardiovascular and musculoskeletal health outcomes, a decrease in psychological well-being, as well as the intention to resign [55]. In this study, we also considered emotional demands and the resulting high emotional load as part of the broader concept of workload. Emotional demands are typical of relational work and refer to exposure to others’ suffering, the need to control or hide one’s feelings in front of supervisors, colleagues or patients/clients, and the need to provide emotional support. In this study, conflicts in terms of values related to the feeling of unaccomplished work and the fear of becoming a vector of the disease due to unsafe working conditions contributed to high emotional demands. The compassion demonstrated by HCWs to patients and their families in the face of suffering and death, especially during strict isolation protocols that kept loved ones apart, was another manifestation of the high emotional demands HCWs experienced during the pandemic, as also documented by Jauvin & Feillou [57]. 

On the contrary, the lack of decisional autonomy did not emerge as a major issue for HCWs during the first two pandemic waves on the social media platforms we explored, which is consistent with the results of other studies [1,2]. It could be that in the context of a health crisis, workers prefer to have quick and clear instructions on how to carry out their tasks and place less emphasis on having a say about how to do the work. This could be different once the crisis abates. 

Other than high workload (including emotional demands), the other main risk factor that emerged from our analyses is the lack of workplace recognition. HCWs can use social media for different purposes, including as a means of sharing experiences with others [58,59,60], of appealing to the population to stay at home or wear a mask, of accessing new evidence and of implementing new practices [59,61]. But HCWs can also use social media in order to seek recognition by promoting themselves professionally and highlighting the importance of nursing [59,62,63]. During the pandemic, attention was given to healthcare professionals through the creation of hashtags such as #heroes, #FrontLineHeroes, #HealtcareHeros or #nowweareheroes [62,64]. As mentioned by Glasdam et al. (2022), the COVID-19 pandemic provided nursing professionals with a unique opportunity to use social media to demonstrate their indispensability. However, such hashtags can also cause backlash, as the hero discourse can be perceived as a way of justifying sacrifice, risk exposure and poor working conditions [62,65]. On 4 April 2020, the Premier of Quebec tweeted “I want to pay tribute to our guardian angels who watch over us and fight this invisible enemy that is COVID-19. I thank you from the bottom of my heart for all you do for us!” [66]. This guardian angel discourse used in Quebec can be considered as a type of hero discourse which, although well-meaning, can have unintended negative consequences, creating frustration. It could explain the feeling of being sacrificed and the lack of recognition that emerged from the Facebook content that was analyzed [65].

Related to the lack of recognition was the feeling of injustice, felt in part as a result of a perceived differential treatment by the media and government of healthcare professions. A study that analyzed reactions to a social media post about who comprises healthcare teams illustrates this point. The post consisted of an image contrasting who society thinks works in hospitals versus who actually works there, with a list of 28 professions in the category of individuals who actually work in hospitals. However, despite the variety of professions listed, the post generated feelings of exclusion, frustration and anger in HCWs, who then pointed out through comments that many professions were omitted [64]. 

In our study, Facebook pages were used to denounce the lack of social support, especially from colleagues or supervisors. These denunciations are important to consider as studies have indicated that the lack of support was one of the main occupational determinants of burnout among HCWs during the COVID-19 pandemic [54,55]. Other studies show that social media could also be used to strengthen the bonds of solidarity between HCWs, such as nurses [67]. Finally, social media can be used in a whistle-blowing capacity to identify detrimental working conditions, such as restrictive working hours or unsafe working conditions due to the lack of PPE [47,50,55,63], and studies indicate that experiencing emotional demands can be fertile ground for calling attention to these matters through the use of different platforms [59,68]. During the pandemic, the inability to use typical channels for dealing with work challenges due to distancing guidelines or the unavailability of supervisors could explain why union Facebook pages were used, allowing HCWs to seek support from peers, build collective solidarity or denounce unsafe working conditions [59,68]. Our study underscores the importance for HCWs to have the possibility to express themselves.

### 4.2. Practical Implications

The identification of risks remains an essential step in an effective intervention process. These results highlight the complexity of the psychosocial work environment and the numerous workplace psychosocial risks to which HCWs reported being exposed through comments made on health union Facebook pages. Monitoring of targeted social media pages could contribute to detecting work-related psychosocial risks using a social media surveillance methodology developed to detect disease outbreaks [69]. This methodology holds the potential for application during a time of major healthcare restructuring and re-organizations.

Without claiming that analysis of social media content is a means of risk assessment, social media, which was used extensively by HCWs during the COVID-19 crisis, can be judiciously used to study work-related issues as expressed by the workers in a timely manner, and to potentially identify targets for prevention. Such monitoring, in combination with other risk assessment data, could improve our understanding of the psychosocial work environment by providing rich information and concrete examples of lived experiences [59]. Given the recent modernization of Quebec’s occupational health and safety regime to explicitly include workplace psychosocial risks [70], and the transition into a post-pandemic world, social media data analysis could be further explored as a tool to follow the long-term impacts of the pandemic and of health-and-safety legislative changes on organizations [71].

### 4.3. Strengths and Limitations

This study has several strengths and limitations. First, this innovative approach allowed us to analyze almost 4000 Facebook comments, yielding rich contextual data without having to solicit HCWs directly. Despite the fact that union pages were targeted, and the potential selection bias towards the most vocal or activist voices, the workplace psychosocial risks identified and the concerns that were expressed on those social media pages are corroborated by findings from epidemiologic studies of Quebec HCWs during the first pandemic waves [1,2,53]. Some professionals, namely doctors and heads of health and social services establishments, may not be reached on social media platforms. But our findings echo those derived from group interviews conducted with Quebec-based doctors and managers over the same time period [57]. Some limitations include non-coverage of those who do not have a Facebook account or who are not likely to express themselves on social media. Data extraction was performed by different individuals using their personal Facebook pages, and despite the use of a standardized extraction form, we cannot be sure to what extent Facebook algorithms influenced the extraction; therefore, it is impossible to guarantee its reproducibility. However, we believe that this is countered by the extensive amount of analyzed comments. Finally, it is important to mention that comments were mostly in reaction to a single post and, as a result, some psychosocial risks may not have been expressed.

## 5. Conclusions

The workplace psychosocial risks to which HCWs reported being exposed in this study of Facebook content, corroborated by findings from other studies, suggest several promising targets for prevention in order to protect HCWs’ mental health and reduce work absences and the intention to resign. Providing HCWs with formal opportunities to express themselves and involving them in the development of interventions to mitigate excessively high workload and the lack of recognition, for example, could yield promising results. Monitoring social media platforms and specifically following targeted pages could be further explored as a means to monitoring work-related psychosocial risks and identifying emerging risks. 

## Figures and Tables

**Figure 1 ijerph-20-06116-f001:**
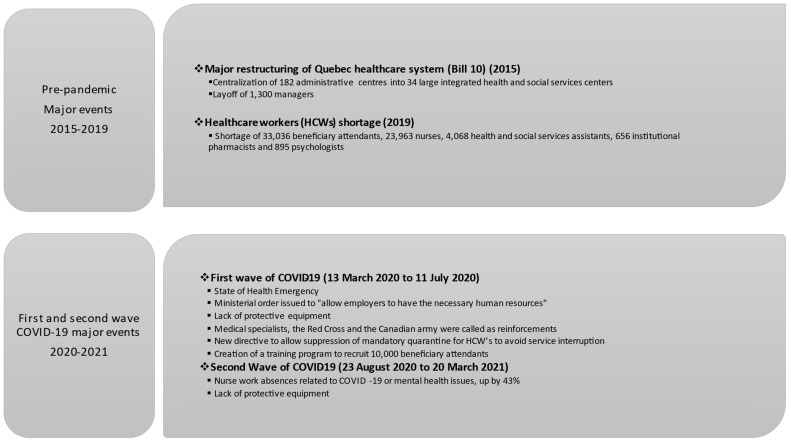
Overview of major events in the Quebec healthcare system prior to and during the COVID-19 pandemic [34,35,36].

**Table 1 ijerph-20-06116-t001:** Work-related psychosocial risk factor definitions and themes.

Work-Related Psychosocial Risk Factor	Definitions [51,52]	Themes
High psychological demands (also known as high workload), including emotional demands.	Excessive amount of work, complexity of the work, time constraints, frequent interruptions and disturbances. Emotional demands refer to the burden experienced given one’s responsibility towards things or people at work or their work roles or mission.	High workload Highly restrictive work hours and schedules; Doubts and uncertainties about scheduling; Mandatory overtime; Cancellation of vacation time; Impossibility to take a break; Lack of staff; Insufficient staff-to-patient ratios; Lack of relief; Welcoming new staff; Staff reassignment; Staff coming from external placement agencies; Emotional demands Value conflicts (not being able to get the work done or endangering patients); The fear of becoming a vector for the virus and infecting patients or loved ones; Compassion for patients suffering or dying.
Low recognition and perceived injustice	Lack of esteem and respect, compensation, job security, or prospects for promotion for efforts and achievements.	Feelings of unfairness and inequitable treatment; Feeling of a lack of respect; Issues with salary; Unfair wages based on the usefulness of work.
Low workplace social support (a) from supervisors/(b) from colleagues	(a) Inability or inaccessibility of the supervisor to offer practical and emotional support to employees; (b) Lack of team spirit, group cohesion, and assistance from colleagues in task completion.	Inadequate support from supervisors (including inadequate informational support); Inadequate support from higher authorities; Inadequate support from unions; Inadequate support from colleagues; Inadequate support from the public.
Work-life conflict	Difficulty balancing obligations towards work and personal life (social activities, care of children or disabled loved ones).	Difficulty balancing work and the care and education of young children during daycare and school closures.
Lack of decisional autonomy	Lack of control over one’s work, through a lack of influence on how to do the work or lack of possibility to use or develop one’s skills and creativity on the job.	Lack of consultation with employees; Insufficient leeway.

## Data Availability

The data are not publicly available due to privacy and ethical restriction.

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
