# Peer review of "Psychosocial Risks among Quebec Healthcare Workers during the COVID-19 Pandemic: A Social Media Analysis"

_ijerph, 2023, doi:10.3390/ijerph20126116_

Round 1

Reviewer 1 Report

Dear authors,

This was a very interesting paper to read. Here are my main, minor recommendations:

1. Correct tenses to reflect that the pandemic is past tense. 

2. In the Introduction, clearly specify why this research is important and how it can be used to address future pandemics, epidemics. For example, on page 2 (lines 88-89), if you looked at what happened during the pandemic, why does it matter now? What use our your findings? You discussed it well in your Discussion, but a brief rationale for the methods you used should be more strongly stated in the Introduction. I think this method is very innovative, and in the Discussion, perhaps point out more uses, such as during restructuring, major technological or practice changes, etc. 

3. In your overview of the pandemic in Quebec, I wonder if you could create some kind of visual timeline? It is so interesting. I also think you should take out the guardian angel quote (p. 3, lines 133+) in this section and put it in the Discussion where you interpret its significance to your findings. 

Minor tense issues. See attachment. 

Reviewer 2 Report

The article “Psychosocial risks among Québec healthcare workers during  the COVID-19 pandemic : A social media analysis ” deal whit an important issue if these experience can provide changes such improve the quality of live of this providers / professional .

Abstract – Clarify that health professionals were at risk of psychosocial risk exposure in Covid -19. Explain who these professionals were and what risks they are referring to (despite the fact that the results are described). Another question refers to the methodology, says the "Media", but FaceBook is a new or traditional  means of social communication like newspapers or television. Use the expression new social media like Facebook social networks.

Introduction - Explain which health professionals you are referring to. Nurses, psychologists, therapists, diagnostic technicians and/or social workers and care assistants. Professionals it's very generic. Perhaps if you related these professionals to the unions that defend them and in which you carried out the study, it would be more clearer.

In the theoretical foundation, authors chooses to describe the health  system, changes and impacts on the provision of health care and on professionals. It would be relevant to continue the literature review here. Use this part to clarify the purpose of the research project.

At the end of this part, he mentions that the research team (page 3 line 146… et seq) started a research project to protect the mental health of professionals whose objective was to build a tool to train them at that level. Maybe this part gets better in the methodology.

Material and methods – Clarify which unions, although everyone could post content. This is important to get an idea of the professionals.

In data extraction – unequivocally demonstrate which grid was made and which variables were adopted in the text extraction. Explain which variables were considered as psychosocial risks.

Examples: themes that were coded were - ¨The main coded themes were high workload, lack of recognition, low social support from colleagues or supervisors, conflicts between work life and lack of decision autonomy.

Would it be relevant to explain the difference between these variables during covid? These are very generic questions and can be appening in health systems even without a pandemic? So, what stood out about the pandemic?

Results: explain the psychosocial risks related to the pandemic, ok.

Discussion of the results: talk about risks to mental health – explain how you reached this conclusion (the discussion has to be associated with the framework that was or was not initially developed). I suggest gounded the framework framework  so that it is possible to relate the results to them)

Authors  mentions that use social media, but he must mention that he used the Facebook channel. So is important not  generalize to all existing social media. You will also be able to reflect on how these social networks can be a way of supporting, denouncing, alerting, and diagnosing these situations, but always in connection with the entities that generate them, which are the unions. People alone cannot change the complexity of the health system and even less through Facebook .

Last comments:

Improve the theoretical foundation and link it to the results. Clarify in detail and visually with tables the methodology and psychosocial categories used for content analysis. As the project focuses on mental health, it is necessary to clarify the concepts is it mental health or psychosocial risks.

There is another issue that is highlighted in the strengths and limitations, is that doctors are not always on Facebook. So who are these people or professionals who made these comments in this pages, this is a weekness.

Clarify the conclusion and respond to the objectives of the study.

Round 2

Reviewer 2 Report

Dear Editor

After observing the answers given and verifying the changes, I believe that the article became clearer in methodological terms and the results obtained. However, I still have some questions, ethical ones, related to the fact of using materials from people on Facebook's social networks to write scientific articles, when we don't know who these people are who make comments on these networks. That's my biggest problem with this article.

I wish good work.